# Stereotactic Radiosurgery for Lung Cancer with a Risk-Adapted Strategy Using the Volumetric Modulated Arc Therapy Technique: A Single Arm Phase II Study

**DOI:** 10.3390/cancers14163993

**Published:** 2022-08-18

**Authors:** Takaya Yamamoto, Yu Katagiri, Yoko Tsukita, Haruo Matsushita, Rei Umezawa, Yoshiyuki Katsuta, Noriyuki Kadoya, Noriyoshi Takahashi, Yu Suzuki, Kazuya Takeda, Keita Kishida, So Omata, Eisaku Miyauchi, Ryota Saito, Keiichi Jingu

**Affiliations:** 1Department of Radiation Oncology, Graduate School of Medicine, Tohoku University, Sendai 980-8574, Japan; 2Department of Respiratory Medicine, Graduate School of Medicine, Tohoku University, Sendai 980-8574, Japan

**Keywords:** stereotactic radiosurgery, stereotactic body radiotherapy, SRS, volumetric modulated arc therapy, lung cancer

## Abstract

**Simple Summary:**

Stereotactic radiosurgery (SRS) for lung cancer has an attractive schedule. In this study, we focused on the efficacy of SRS, and the primary endpoint of this study was the 3-year local recurrence rate. The results showed that the 3-year local recurrence rate was 5.3% (95% confidence interval: 0.3–22.2%), and this rate was less than the expected rate. Good results were obtained in this study and this regimen of SRS is a candidate for a future phase III trial.

**Abstract:**

Purpose: A phase II study carried out to assess the efficacy of a risk-adapted strategy of stereotactic radiosurgery (SRS) for lung cancer. The primary endpoint was 3-year local recurrence, and the secondary endpoints were overall survival (OS), disease-free survival (DFS), rate of start of systemic therapy or best supportive care (SST-BSC), and toxicity. Materials and Methods: Eligible patients fulfilled the following criteria: performance status of 2 or less, forced expiratory volume in 1 s of 700 mL or more, and tumor not located in central or attached to the chest wall. Twenty-eight Gy was prescribed for primary lung cancers with diameters of 3 cm or less and 30 Gy was prescribed for primary lung cancers with diameters of 3.1–5.0 cm or solitary metastatic lung cancer diameters of 5 cm or less. Results: Twenty-one patients were analyzed. The patients included 7 patients with adenocarcinoma, 2 patients with squamous cell carcinoma, 1 patient with metastasis, and 11 patients with clinical diagnosis. The median tumor diameter was 1.9 cm. SRS was prescribed at 28 Gy for 18 tumors and 30 Gy for 3 tumors. During the median follow-up period of 38.9 months for survivors, 1 patient had local recurrence, 7 patients had regional or distant metastasis, and 5 patients died. The 3-year local recurrence, SST-BSC, DFS, and OS rates were 5.3% (95% confidence interval [CI]: 0.3–22.2%), 20.1% (95% CI: 6.0–40.2%), 59.2% (95% CI: 34.4–77.3%), and 78.2% (95% CI: 51.4–91.3%), respectively. The 95% CI upper value of local recurrence was lower than the null local recurrence probability. There was no severe toxicity, and grade 2 radiation pneumonitis occurred in 1 patient. Conclusions: Patients who received SRS for lung cancer had a low rate of 3-year local recurrence and tolerable toxicity.

## 1. Introduction

Patients who have undergone stereotactic body radiotherapy (SBRT) for early stage lung cancer have shown higher local control and overall survival (OS) rates than those for patients who have received conventional fractionated radiotherapy [1]. In addition to this superiority of SBRT, the treatment time for SBRT is much shorter with only a single-fraction schedule required for so-called stereotactic radiosurgery (SRS). Although the outcomes of SRS for early stage lung cancer or metastatic lung cancer have gradually been accumulating, the number of institutes in which SRS is performed is small [2,3,4]. In the RTOG 0915 trial in which 34 Gy in 1 fraction was compared to 48 Gy in four fractions for early stage lung cancer, the 34-Gy SRS arm showed lower rates of severe toxicities with primary control rates comparable to those of the 48-Gy SBRT arm [5]. Despite these results, 48 Gy in four fractions has been the most frequent dose prescription in Japan [6]. One of the reasons is that many radiation oncologists did not have knowledge about the optimal dose of SRS. In the RTOG 0915 trial, 34 Gy resulted in 97% primary tumor control at 1 year, but another study showed that 1-year local recurrence rates were 13.8% in patients who received 34-Gy SRS and 2.0% in patients who received 30-Gy SRS [7]. Lower radiation doses such as 26-Gy SRS also resulted in a high local control rate [8]. In lung SRS, a higher radiation dose might not always provide better treatment outcomes. We therefore started a prospective single-arm SRS study for lung cancer with a risk-adapted strategy using the volumetric modulated arc therapy (VMAT) technique in 2016 [9]. The risk-adapted strategy was that SRS doses were changed according to the risks of local recurrence. Patients were divided into two risk groups based on previous results of predictive factor analyses for local control [10]. A normal risk group was primary lung cancers with diameters of 3 cm or less and a high risk group was primary lung cancers with diameters of 3.1–5.0 cm or metastatic lung cancer with diameters of 5 cm or less. Radiation dose for the normal risk group was 28 Gy, which was determined by the results of a phase I SRS-VMAT study and was almost the same biological effective dose (BED_10_) as 48 Gy in four fractions [11]. Radiation dose for the high risk group was increased up to 30 Gy [7]. The primary endpoint of this prospective phase II study was 3-year local recurrence, and the secondary endpoints were OS, disease-free survival (DFS), percentage of patients who received systemic therapy or best supportive care (SST-BSC), and toxicity. In the exploratory analysis, SRS was compared with multi-fraction SBRT using propensity score matching.

## 2. Materials and Methods

### 2.1. Eligibility Criteria and Informed Consent

Candidates were patients who were medically inoperable or who refused surgery for primary lung cancer or solitary metastatic lung cancer with diameters of 5.0 cm or less and were indicated for SBRT. Eligible patients fulfilled the following criteria: performance status of 2 or less, forced expiratory volume in 1 s of 700 mL or more, and eligible lung tumor not located in central or not attached to the chest wall. Patients who needed oxygen inhalation, patients with a past history of thoracic radiotherapy or pneumonectomy, and patients with an interstitial shadow on CT images were excluded. Written informed consent was obtained from all patients. This study was performed at Tohoku University Hospital and was approved by the Ethical Committee of Tohoku University Hospital on July 15, 2016 (reference number: 2016-2-086-1). Patients were recruited between July 2016 and June 2020 and the date of data cut-off was 25 June 2022.

### 2.2. SRS Procedure

Radiotherapy planning CT scans at intervals of 2.0–2.5 mm and a 4-dimensional CT scan of the whole lung were performed with or without respiratory motion control using an abdominal pressure system and/or oxygen inhalation if needed. An abdominal compression paddle (Qfix, Avondale, PA, USA) or a pneumatic abdominal compression belt (FREEDOM compression belt, CDRSystems, Calgary, Canada) were used for respiratory motion control. Tumor and organ delineation and radiotherapy planning were performed using Eclipse (Varian Medical Systems, Palo Alto, CA, USA). Gross tumor volume (GTV) was contoured on the basis of the visible extent of the tumor on planning CT, and internal GTV was contoured by using 4-D CT images. Planning target volume (PTV) was created by adding 5 mm to all directions around the internal GTV.

The radiotherapy technique was VMAT with 6 MV X-ray beams, and Acuros XB was used for the calculation algorithm. VMAT was performed with a linear accelerator (Clinac 23EX or TrueBeam STx, Varian Medical Systems, Palo Alto, CA, USA). For primary lung cancers with diameters of 3 cm or less (a normal risk group), 28 Gy in 1 fraction was prescribed for covering 95% of the PTV (D95), 98% of the PTV (D98) exceeded 26 Gy, and 2% of the PTV (D2) exceeded 34–36 Gy with recommendation of 36 Gy after publication of a tumor control probability model [12]. For primary lung cancers with diameters of 3.1–5.0 cm or metastatic lung cancer with diameters of 5 cm or less (a high risk group), 30 Gy in 1 fraction was prescribed to D95, D98 exceeded 27 Gy, and D2 exceeded 38 Gy. Dose constraints for surrounding organs are shown in the Appendix A.

### 2.3. Follow-Up, Outcome Definitions, and Sample Size

A clinical examination and CT scanning were performed 4–8 weeks after SRS. Thereafter, patients underwent CT scans every 6 months for 3 years. When it was difficult to distinguish local recurrence from radiation pneumonitis or radiation fibrosis, an additional FDG-PET examination or shorter follow-up of CT scanning was recommended.

Local recurrence was defined as recurrence of the irradiated lung tumor. DFS was defined as freedom from any recurrence, metastasis, other cancer including second primary cancer, or death. OS was defined as freedom from death from any cause. SST-BSC rate was defined as percentage of patients who received systemic therapy or best supportive/palliative care because of recurrent or metastatic primary disease that was not salvaged by local therapy. Toxicity was judged according to the National Cancer Institute Common Terminology Criteria for Adverse Events version 5.0 (CTCAE v5.0).

According to our previous data, the 3-year cumulative incidence of local recurrence was 22.6% (95% confidence interval [CI]: 17.3–28.4%) (Appendix A) [11]. Local control rate of SRS was thought to be around 10% [2]. Therefore, we set the expected local recurrence probability, null local recurrence probability, alpha and beta as 10.0%, 28.4%, 0.05 and 0.3, respectively [13]. The calculated sample size was 21 patients (1-sided), and 22 patients were enrolled with consideration of withdrawal from this study or lost to follow-up.

### 2.4. Statistical Analyses

All time-to-event data were calculated from the date when SRS was performed to the date when the event was confirmed. Then the rate of each event was estimated by using the Kaplan–Meier method. When the cumulative incidence of local recurrence, SST-BSC or radiation pneumonitis was calculated, death was regarded as a competing event. In the exploratory analyses, SRS was compared with multi-fraction SBRT, which has been used in clinical practice in our institute. Multi-fraction SBRT consisted of 4- and 8-fraction SBRT, and 48 Gy in four fractions or 60 Gy in 8 fractions were typically prescribed to D95. Patients who were treated with multi-fraction SBRT for primary or metastatic lung cancer in the same period as that of this study and whose follow-up period was more than 6 months were identified from our database. Then, the propensity score was estimated by logistic regression using the following covariates: age, sex, performance status, Charlson comorbidity index, diagnosis (pathologically diagnosed primary lung cancer, clinically diagnosed primary lung cancer or metastatic lung cancer), tumor diameter, and interstitial shadow. One-to-one matching was performed using nearest neighbor matching within a caliper width that was equal to 0.25 of the standard deviation of the logit of the propensity score [14]. Absolute standardized differences of less than 0.25 were regarded as successful matching [15]. Then, survival data were compared using the log-rank test and categorical data were tested using Fisher’s exact test. Sensitivity analysis was also performed by inverse probability of treatment weighting using the same covariates. A *p*-value less than 0.05 was defined as significant. Statistical analyses were performed using EZR version 1.54 (Saitama Medical Center, Jichi Medical University, Saitama, Japan), a modified version of R commander (R Foundation for Statistical Computing, Vienna, Austria) [16].

## 3. Results

A total of 22 patients were enrolled, but 1 patient withdrew from the study before the treatment. Therefore, 21 patients were analyzed, and the characteristics of those patients are shown in Table 1. All of those patients met the eligibility criteria and completed radiotherapy. The results of radiotherapy planning had no protocol dose violation. Additional systemic therapy was not performed until confirmation of disease progression.

The median follow-up period for all patients was 36.2 months (range: 12.4–66.3 months) and that for survivors was 38.9 months (range: 12.4–66.3 months). At the time of data cut-off, a total of 5 patients had died with a median interval of 26.6 months (range: 13.0–47.7 months) and 1 patient among them died without evidence of any recurrence. There was 1 case of local recurrence at 27 months after SRS, and SBRT was performed again for the local recurrence site. Regional or distant metastasis occurred in 7 patients: regional lymph node metastases in 2 patients, multiple brain metastasis in 1 patient, liver metastases in 1 patient who were treated with SRS for a lung metastasis from liver cancer, multiple lung metastases in 1 patient, and second primary lung cancer or solitary lung metastasis in two patients. The last two patients received another lung SBRT for a newly emerged lesion. The 3-year cumulative incidence rates of local recurrence and SST-BSC were 5.3% (95% CI: 0.3–22.2%) and 20.1% (95% CI: 6.0–40.2%), respectively (Figure 1). The 95% CI upper value of local recurrence rate of 22.2% was lower than the null local recurrence probability of 28.4%. The estimated 3-year DFS rate and OS rate were 59.2% (95% CI: 34.4–77.3%) and 78.2% (95% CI: 51.4–91.3%), respectively (Figure 2). Regarding the toxicity of radiotherapy, grade 2 radiation pneumonitis occurred in 1 patient in 28-Gy arm and grade 3 or higher radiation pneumonitis did not occur. The 3-year cumulative incidence of grade 2 or higher radiation pneumonitis was 4.8% (95% CI: 0.3–20.2%). Radiation-induced rib fracture occurred only in 28-Gy arm: 6 patients with grade 1 and 2 patients with grade 2.

Propensity score matching was applied for the SRS cohort and multi-fraction SBRT cohort, and 36 patients were identified. Characteristics of each cohort before and after matching are shown in Appendix A. Before propensity score matching, the 3-year local recurrence, SST-BSC, DFS, and OS rates for the multi-fraction SBRT cohort were 12.0% (95% CI: 5.2–22.2%, *p* = 0.37), 35.0% (95% CI: 22.8–47.3%, *p* = 0.07), 45.1% (95% CI: 31.7–57.5%, *p* = 0.06), and 68.3% (95% CI: 53.2–79.4%, *p* = 0.58), respectively (Figure 3). After the matching, local recurrence, SST-BSC, DFS, and OS rates for SRS matched cohort were 6.3% (95% CI: 0.4–25.8%), 17.8% (95% CI: 4.1–39.5%), 63.8% (95% CI: 36.1–82.1%), and 80.1% (95% CI: 49.6–93.3%), respectively, and those for the multi-fraction SBRT matched cohort were 15.2% (95% CI: 2.2–39.5%, *p* = 0.45), 40.7% (95% CI: 17.1–63.3%, *p* = 0.03), 59.3% (95% CI: 32.5–78.4%, *p* = 0.29), and 75.8% (95% CI: 46.9–90.3%, *p* = 0.89), respectively (Figure 3). Rib fracture occurred in 5 patients with grade 1 and 2 patients with grade 2 in the SRS cohort but only 1 patient with grade 1 in the multi-fraction SBRT cohort (*p* = 0.04). There was no significant difference between the two cohorts in the rate of grade 2 or more radiation pneumonitis (5.6% vs. 0.0%, *p* = 0.31). In the sensitivity analysis, there were no significant differences in local recurrence, SST-BSC, DFS and OS (*p* = 0.51, *p* = 0.35, *p* = 0.25 and *p* = 0.79, respectively).

In this study, a low rate of local recurrence was achieved by using moderate SRS doses with tolerable toxicities. Risk-adapted radiation doses with the VMAT technique would also work well. Reports of SRS for lung cancer such as outcomes in the RTOG 0915 trial have gradually been accumulating, but the rate of adoption of this schedule for SBRT has been relatively low. There are some possible reasons for this. First, the rates of OS in the 34-Gy arm in the RTOG 0915 trial were relatively low: 2-year OS rates in the 34-Gy SRS arm and 48-Gy SBRT arm were 61.3% and 77.7%, respectively. Second, some radiation oncologists put emphasis on the reoxygenation phenomenon, and some institutions have therefore preferred multi-fraction SBRT. For example, a radiobiology-based regimen of four-fraction SBRT for lung cancer with intervals of at least 72 h has been reported [17]. Third, since a phase III nationwide clinical trial of four-fraction SBRT for primary lung cancer is ongoing in Japan, SRS has not been introduced in many institutes [18]. Finally, optimal SRS doses have not been established because of the lack of results of a phase III trial.

There has been discussion of optimal SRS doses. In the RTOG 0915 trial, an SRS dose of 34 Gy was selected on the basis of previous findings for SRS performed between 1998 and 2004 [19]. However, the dose at that time was different from doses used in recent trials because of differences in the dose calculation algorithms, especially heterogeneity correction, and target coverage of the prescribed dose [20]. Furthermore, a previous study showed a higher recurrence rate in patients who received 34-Gy SRS than in patients who received 30-Gy SRS [7]. It was thought that a much higher dose might lead to inadequate GTV delineation, an inadequate PTV margin, or the need for a complex radiation treatment plan because of anxiety about increased toxicity. Therefore, 28 Gy and 30 Gy were selected in this study and patients treated with these doses showed local recurrence rates comparable to those in a previous study on SRS [2]. Afterwards, that report comparing 30-Gy SRS with 34-Gy SRS was updated, and there was no difference between local recurrence rates in patients who received 30-Gy SRS and patients who received 34-Gy SRS [21]. In another study, the 3-year local progression-free survival rate in patients who received 30-Gy SRS was 87.8% and it was not significantly different from that in patients who received 70 Gy in 10 fractions [22]. As for 28-Gy SRS, it was reported that there was no difference between the outcomes for patients who received 28-Gy SRS and 48 Gy in four fractions for lung oligometastases [23]. In the present study, patients who received 28- or 30-Gy SRS also had a low local recurrence rate, and this would be because of the precise radiation planning requirement that was based on a tumor control probability model [12]. As a result, the median doses of mean internal GTV doses in the 28-Gy arm and 30-Gy arm were 35.5 Gy and 40.1 Gy, respectively.

Our primary endpoint was 3-year local recurrence because previous data showed that local recurrence rate reached near plateau around 3 years (Appendix A). Although there has been a gradual accumulation of evidence for the effectiveness of SRS, there have been few reports on the long-term outcomes of SRS. In the RTOG 0915 trial, the 1-year primary tumor control rate of 34-Gy SRS arm was 97.0%, and long-term follow-up data showed a 5-year primary tumor recurrence rate of 10.6% [5,24]. In the TROG 13.01 trial, 1- and 3-year freedom from local recurrence rates in the 28-Gy SRS arm were 93% and 64%, respectively [23]. Although the primary endpoint of both trials was safety, these results suggested that a relatively long follow-up period was needed to determine efficacy. In the present study, only 1 of the 21 patients developed local recurrence during the median follow-up period of 38.9 months, and the 3-year local recurrence rate was only 5.3% with a 95% CI upper value of 22.2%, which is lower than the 95% CI upper value of 28.4% in previous local recurrence data and even slightly lower than previously reported 3-year local recurrence rate of 22.6%.

Another advantage of SRS is that it provides a steeper dose distribution than that of multi-fraction SBRT even in the same VMAT plan. Alongi et al. compared two radiation schedules based on almost the same BED_10_: 70 Gy in 10 fractions of BED_10_ 119 Gy and 30 Gy in 1 fraction of BED_10_ 120 Gy [23]. When 150% of the prescribed dose was delivered to the GTV, the BED_10_ values were 215 Gy and 247 Gy, respectively. On the other hand, when the adjacent organ received 50% of the prescribed dose, the BED_10_ values were 47.2 Gy and 37.5 Gy, respectively. Although caution is needed because the BED_3_ values were 75.8 Gy and 90 Gy, respectively, this feature of BED is one of the strengths of SRS.

The SST-BSC rate after SRS might be lower than that after multi-fraction SBRT. Although sensitivity analysis showed no significance, there were differences between SST-BSC rates in patients who received SRS and patients who received multi-fraction SBRT before and after propensity score matching (Figure 3). There was no difference in progression-free survival rates between SRS and multi-fraction SBRT in some trials [23,24,25]. However, patients who received SRS for primary renal cell carcinoma showed higher distant control, progression-free survival, and cancer-specific survival rates than those in patients who received multi-fraction SBRT, despite the fact that local control rates were almost the same [26]. In the treatment of oligometastatic cancer, patients who received 24-Gy SRS showed lower rates of local recurrence and distant metastasis than those in patients who received three-fraction SBRT [27]. Analyses of immunogenic effects in the TROG 13.01 trial showed increases in T-regulatory cells, cytotoxic T-lymphocyte–associated antigen and programmed cell death 1 expression after SRS or SBRT with slight differences between SRS and four-fraction SBRT [23]. There is a possibility that SRS has an advantage of a systemic effect.

Regarding toxicity, there was no severe adverse event after SRS and only 1 patient developed grade 2 radiation pneumonitis. On the other hand, the rate of radiation-induced rib fracture was higher in patients who received SRS than in patients who received multi-fraction SBRT. The rate was high considering that no attachment of the lung tumor to the chest wall was one of the eligibility criteria of this study and dose constraints of rib or chest wall were not used in multi-fraction SBRT. A similar difference was seen in the RTOG 0915 trial: 7 of the 39 patients in the 34-Gy SRS arm had injuries including fractures, whereas only 1 of the 45 patients in the 48-Gy SBRT arm had an injury [5]. Although maximum dose of rib was limited to 30 Gy or less in this study, additional and more strict dose constraints of rib would be desirable. In the review article, the volume of chest wall and ribs receiving 22 Gy or more and 28 Gy or more were limited to 1 cc or less and 5 cc or less, respectively [28]. In a previous study, there was a relatively high rate of rib fracture (in 41 of 177 patients) and it was shown that a tumor-chest wall distance and female sex were risk factors for rib fracture [29]. Although VMAT has the advantage of chest wall dose reduction, our prescription method of D98 exceeding 26 Gy might result in a higher rate of rib fracture [30].

There are some limitations in this study. This prospective study was performed in a single institute with a small sample size design: single arm, relatively low power, and one-sided statistical design. Many patients with no pathological confirmation were included. Furthermore, 30-Gy SRS arm was only 3 patients compared to 18 patients for 28-Gy SRS arm. This is the problem of the study design; therefore, a reasonable study design is needed. Stratification according to the risks might be useful [31]. A future prospective randomized controlled phase III design and trial are needed.

## 4. Conclusions

In conclusion, SRS for lung cancer is effective and the outcome is comparable to that of multi-fraction SBRT in the long term. Although the rate of radiation-induced rib fracture was high, there was no severe adverse event and the rate of grade 2 lung toxicity was only 4.7%. The 3-year cumulative incidence rate of local recurrence, which was the primary endpoint of this study, was 5.3% (95% CI: 0.3–22.2%). The 95% CI upper value of local recurrence is lower than the null local recurrence probability. Therefore, this risk-adapted strategy for SRS is appropriate and this regimen is a candidate for a future phase III trial.

## Figures and Tables

**Figure 1 cancers-14-03993-f001:**
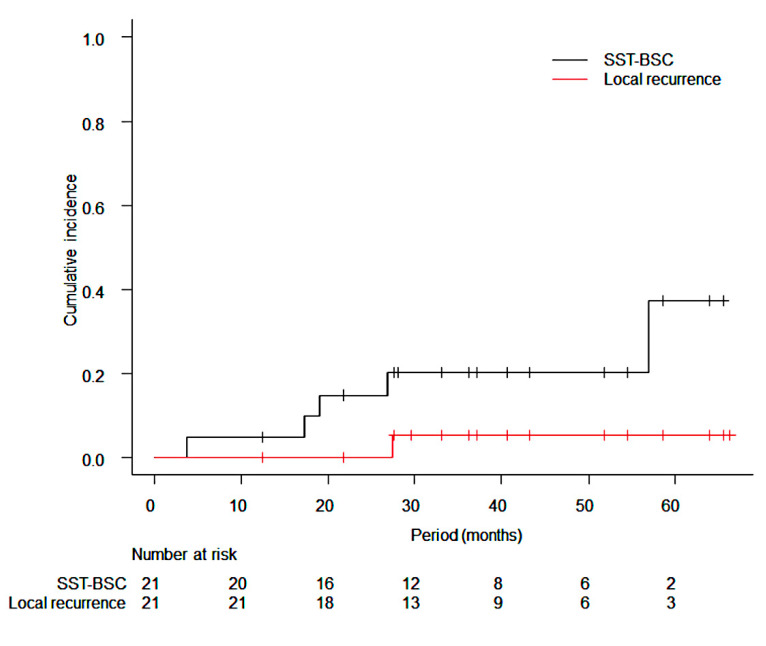
Cumulative incidence rates of local recurrence and percentage of patients who received systemic therapy or best supportive/palliative care (SST-BSC). The 1-year, 2-year and 3-year local recurrence rates were 0.0%, 0.0%, and 5.3%, respectively. The 1-year, 2-year, and 3-year SST-BSC rates were 4.8%, 14.8%, and 20.1%, respectively.

**Figure 2 cancers-14-03993-f002:**
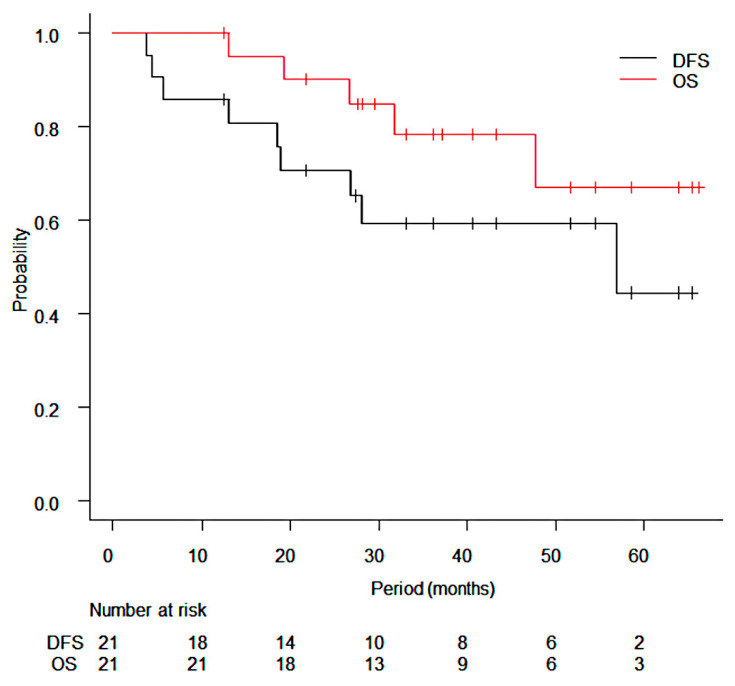
Kaplan–Meier curves of overall survival (OS) and disease-free survival (DFS). The 1-year, 2-year, and 3-year OS rates were 100.0%, 90.0%, and 78.2%, respectively. The 1-year, 2-year, and 3-year DFS rates were 85.7%%, 70.6%, and 59.2%, respectively.

**Figure 3 cancers-14-03993-f003:**
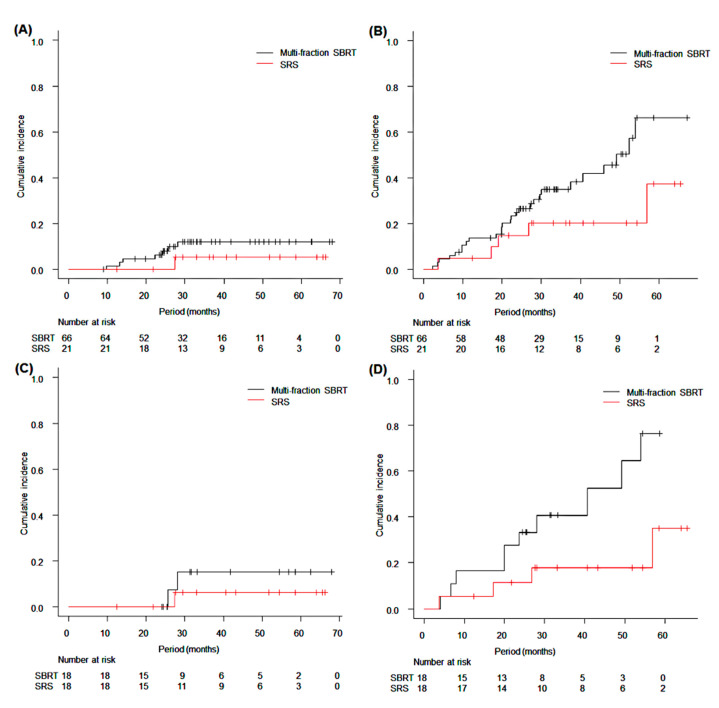
(**A**) Curves of cumulative incidences of local recurrence after stereotactic radiosurgery (SRS) and after multi-fraction stereotactic body radiotherapy (SBRT) before propensity score matching. (**B**) Curves of cumulative incidences of percentage of patients who received systemic therapy or best supportive/palliative care (SST-BSC) after SRS and after multi-fraction SBRT before propensity score matching. (**C**) Curves of cumulative incidences of local recurrence after SRS and after multi-fraction SBRT after propensity score matching. (**D**) Curves of cumulative incidence of SST-BSC rate after SRS and after multi-fraction SBRT after propensity score matching.

**Table 1 cancers-14-03993-t001:** Patient characteristics.

Category	Variables	21 Patients (%)
Age, years	Median (range)	74 (59–83)
Sex	Female	3 (14.2%)
	Male	18 (85.7%)
Performance status	0	9 (42.8%)
	1	10 (47.6%)
	2	2 (9.5%)
Charlson comorbidity index	0–1	5 (23.8%)
	2	8 (38.0%)
	3	4 (19.0%)
	4–5	4 (19.0%)
FEV1 (L)	Median (range)	2.2 (0.8–3.5)
FEV1 (% of predicted)	Median (range)	85.2 (28.3–123.6)
Pathology	Adenocarcinoma	7 (33.3%)
	Squamous cell carcinoma	2 (9.5%)
	Metastasis	1 (4.7%)
	Clinical diagnosis	11 (52.3%)
Tumor diameter, cm	Median (range)	1.9 (1.0–3.2)
Radiation dose, Gy	28 Gy	18 (85.7%)
	30 Gy	3 (14.2%)
Mean internal GTV dose, Gy	Median (range) of 28-Gy arm	35.5 (32.7–39.5)
	Median (range) of 30-Gy arm	40.1 (38.8–40.3)

Abbreviations: FEV1: forced expiratory volume in 1 s, GTV: gross tumor volume.

## Data Availability

The data presented in this study are available on request from the corresponding author. The data are not publicly available due to the inclusion of unpublished data.

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
