# Peer review of "Stereotactic Radiosurgery for Lung Cancer with a Risk-Adapted Strategy Using the Volumetric Modulated Arc Therapy Technique: A Single Arm Phase II Study"

_cancers, 2022, doi:10.3390/cancers14163993_

Round 1

Reviewer 1 Report

Dear Author,

you present your phase II study about stereotactic radiosurgery for lung cancer, using the volumetric modulated arc therapy technique for minimizing post-operative risks.

The authors take into account an original population of 22 patents; they finally analyzed operative and post-operative outcomes of 21 patients.

Results are satisfactory in terms of feasibility and safety of the technique. Only 1 patientes developped radiation pneumonitis. Nevertheless you founded rib fractures in 7 patients treated by SRS, compared with the only patient with rib fracture treated by SBRT.

It is your intention to develop some tricks on the protocol to minimize this complication? 

An other remark: I think that pre operative pathological confirmation and histological diagnosis is mandatory, even for the future therapeutic possibilities of the patients (possible biologic or target therapy if adenocarcinoma, or immunotherapy if PD-L1 mutation) in case of relapse.

I agree with the authors about the other limitations of the study. the sample size is small and performed in a single institute. In addition there is a single arm of patients.

However I think that this risk-adapted strategy for SRS may be appropriate with some modification, if possible, in order to minimize adverse events.

Author Response

Response for Reviewer 1

We are grateful for your comments and advices. We revised our manuscript and your advices enabled us to improve the quality of our manuscript.

# English language and style are fine/minor spell check required

Response:

We reviewed our manuscript and corrected some misspelling and spaces.

# Results are satisfactory in terms of feasibility and safety of the technique. Only 1 patientes developped radiation pneumonitis. Nevertheless you founded rib fractures in 7 patients treated by SRS, compared with the only patient with rib fracture treated by SBRT.

It is your intention to develop some tricks on the protocol to minimize this complication?

Response:

Thank you very much for your comment and question. Higher rate of rib fractures in SRS was an unexpected result. This was because tumor location and dose constraints. In this study, maximum dose of rib was limited to 30 Gy or less, on the other hand, dose constraints of rib or chest wall were not used in multi-fraction SBRT. We thought that additional dose constraints of rib was needed. We added these things to discussion section.

“ The rate was high considering that no attachment of the lung tumor to the chest wall was one of the eligibility criteria of this studyand dose constraints of rib or chest wall were not used in multi-fraction SBRT. A similar difference was seen in the RTOG 0915 trial: 7 of the 39 patients in the 34-Gy SRS arm had injuries including fractures, whereas only 1 of the 45 patients in the 48-Gy SBRT arm had an injury [5]. Although maximum dose of rib was limited to 30 Gy or less in this study, additional and more strict dose constrain of rib was desirable. The volume of chest wall and ribs receiving 22 Gy or more and 28 Gy or more were limited to 1 cc or less and 5 cc or less, respectively in the review article [28].”

 (The reason for no dose constraint of rib in multi-fraction SBRT was that our dose constraints in clinical practice was based on JCOG1408, a phase III trial in Japan. There was no dose constraint of rib or chest wall in JCOG1408 trial. reference number18 in the manuscript and constraints were attached).

# An other remark: I think that pre operative pathological confirmation and histological diagnosis is mandatory, even for the future therapeutic possibilities of the patients (possible biologic or target therapy if adenocarcinoma, or immunotherapy if PD-L1 mutation) in case of relapse.

Response:

Thank you for your comments. We agree with you, and almost all patients was tried to obtain pathological confirmation by transbronchial lung biopsy (TBLB). But, the success rate was not high because the lung tumor was relatively small. If pathological confirmation was not obtained, we could choice another modality, a CT-guided biopsy. But, because CT-guided biopsy had some risks such as pneumothorax and hemorrhage, we often diagnosed clinically lung cancer. After the recurrence, we chose adequate modality for pathological confirmation according to recurrence sites: TBLB, endobronchial ultrasound guided transbronchial needle aspiration (EBUS-TBNA), CT-guided biopsy or surgical resection. Therefore, clinical diagnosis cases were eligible in this study.

Sadly, one patient suffered cerebral infarction because of cessation of anticoagulant for preoperative biopsy, then he participated in this study.

#I agree with the authors about the other limitations of the study. the sample size is small and performed in a single institute. In addition there is a single arm of patients.

However I think that this risk-adapted strategy for SRS may be appropriate with some modification, if possible, in order to minimize adverse events.

Response:

Thank you very much for your encouraging comments. We will use this experience for future trials.

Reviewer 2 Report

The study of SRS for the lungs is rare and could be very useful as a paper.

I have a few questions. They are listed below.

1.       Abstract

The 3-year local recurrence, SST-BSC, DFS and OS rates were 5.3% (95% confidence interval [CI]: 0.3-22.2%), 20.1% (95% CI: 6.0-40.2%), 59.2% (95% CI: 34.4-77.3%) and 78.2% (95% CI: 51.4-91.3%), respectively.

Q. The 3-year local recurrence rate is too low compared to the low DFS.

Are there exclusion criteria?

2.       Materials and Methods-SRS procedure-

Radiotherapy planning CT scans at intervals of 2.0-2.5 mm and a 4-dimensional CT scan of the whole lung were performed with or without respiratory motion control using an abdominal pressure system and/or oxygen inhalation if needed.

Q. Coulud you describe the respiratory motion control system specifically?

Why is it not done with breath-holding system?

If VMAT involves respiratory movement, does it not involve uncertainty?

3.       Discussion

First, the rates of OS in the 34-Gy arm in the RTOG 0915 trial were relatively low: 2-year OS rates in the 34-Gy SRS arm and 48-Gy SBRT arm were 61.3% and 77.7%, respectively.

Q. What is the reason for the improved OS in this study compared to previous studies?

Did you increase the dose inside the tumor?

You mention that the primary endpoint was 3-year local recurrence because previous data showed that the local recurrence rate reached a plateau around 3 years (Fig. S1), but we would like to know the results at 5 years.

Author Response

Response for Reviewer 2

We are grateful for your comments and advices. We revised our manuscript and your advices enabled us to improve the quality of our manuscript.

  1. Abstract

The 3-year local recurrence, SST-BSC, DFS and OS rates were 5.3% (95% confidence interval [CI]: 0.3-22.2%), 20.1% (95% CI: 6.0-40.2%), 59.2% (95% CI: 34.4-77.3%) and 78.2% (95% CI: 51.4-91.3%), respectively.

  1. The 3-year local recurrence rate is too low compared to the low DFS.

Are there exclusion criteria?

Response:

Thank you for your question. As you pointed out, there was only 1 local recurrence in contrast to 9 events of DFS. The discrepancy was thought to come from inclusion criteria and the definition of DFS which was freedom from any recurrence, metastasis, other cancer including second primary cancer or death.

The details of 9 events was as follows: local recurrence in 1 patient, regional lymph node metastases in 2 patients, multiple brain metastasis in 1 patient, liver metastases in 1 patient who were treated with SRS for a lung metastasis from liver cancer, multiple lung metastases in 1 patient, second primary lung cancer or solitary lung metastasis in 2 patients and 1 patient died without evidence of any recurrence. We added recurrence information in result section.

Exclusion criteria of this study was that patients who needed oxygen inhalation, patients with a past history of thoracic radiotherapy or pneumonectomy, and patients with a interstitial shadow on CT images.

  1. Materials and Methods-SRS procedure-

Radiotherapy planning CT scans at intervals of 2.0-2.5 mm and a 4-dimensional CT scan of the whole lung were performed with or without respiratory motion control using an abdominal pressure system and/or oxygen inhalation if needed.

  1. Coulud you describe the respiratory motion control system specifically?

Response:

Thank you for your advice. We added the following sentence:

Abdominal compression paddle (Qfix, Avondale, PA) or pneumatic abdominal compression belt (FREEDOM compression belt, CDRSystems, Calgary, Canada) were used for respiratory motion control.

  1. Why is it not done with breath-holding system?

Response:

Thank you for your question. Breath-holding technique have some advantages, but breath-holding was not used in this study. We thought that long treatment time would be needed to perform SRS. Actually, the Monitor-Unit (MU) values sometimes exceeded 10000 MU in SRS. Furthermore, treatment machine was varian 23EX or turebeam STx for SRS, and maximum dose rate of 23EX was 600 MU/min. If we used breath-holding system, patients were needed to perform breath hold many times. We were afraid that breath-hold might be inaccurate in some patients in such a long treatment time. Therefore, we chose abdominal pressure system in this study.

The information of treatment machine was important, and we added the information in the manuscript.

  1. If VMAT involves respiratory movement, does it not involve uncertainty?

Response:

Thank you for your question. Yes, VMAT involves respiratory movement, therefore, we used 4-D CT images to create internal GTV. Furthermore, it involves another uncertainty which is called inter-play effect. Abdominal pressure system did not always improve lung dose-volume parameter because the volume of lung was decreased by abdominal compression. But, abdominal pressure system was important and was used to minimize the effect of inter-play effect.

  1. Discussion

First, the rates of OS in the 34-Gy arm in the RTOG 0915 trial were relatively low: 2-year OS rates in the 34-Gy SRS arm and 48-Gy SBRT arm were 61.3% and 77.7%, respectively.

  1. What is the reason for the improved OS in this study compared to previous studies?

Response:

Thank you for your question. The 3-year OS was 78.2% in this study, and the rate was slightly better 48-Gy arm in RTOG 0915. One of the reason was that the patients in RTOG 0915 trial was medically inoperable non-small cell lung cancer patients, on the other hand, this study also included operable patients who refused surgery. Because OS was secondary endpoint in this study and RTOG 0915, we thought these difference were not significant findings.

  1. Did you increase the dose inside the tumor?

Response:

Yes, we did. In 28-Gy arm, 2% of the PTV should exceed 34-36 Gy and 2% of the PTV should exceed 38 Gy in 30-Gy arm. Therefore, the median dose of mean internal GTV dose were 35.5 Gy and 40.1 Gy, respectively (Table 1).

You mention that the primary endpoint was 3-year local recurrence because previous data showed that the local recurrence rate reached a plateau around 3 years (Fig. S1), but we would like to know the results at 5 years.

Response:

Thank you for your comments. The 3-year and 5-year cumulative incidence of local recurrence was 22.6% and 25.3%, respectively. This was thought to be “near” plateau, therefore we modified the expression.

Reviewer 3 Report

In this study the authors reported the results of a single institute phase II study carried out to assess the efficacy of a stereotactic radiosurgery (SRS) scheme for lung cancer. Based on the data of 21 patients they concluded that patients received SRS had a low rate of 3-year local recurrence and tolerable toxicity.

The manuscript was written concisely and contains information potentially useful. However, there are several of issues to be addressed clearly.

1) Please describe in detail what the risk-adapted strategy was and rationale underneath it.

2) There were only 3 patients for 30 Gy prescription compared to 18 patients for 28 Gy prescription, is it reasonable to analyze all together?  Provide your justification and discussion about it.  In addition, you may want to talk about whether complicated cases were from high dose group or not.

Specific Comments   

P2: Regarding “Primary lung cancers with diameters of 3 cm or less were treated with 28-Gy SRS and primary lung cancers with diameters of 3.1-5.0 cm or metastatic lung cancer were treated with 30-Gy SRS” was it for “primary” only? This conflicts with “Patients who were medically inoperable or who refused surgery for primary lung cancer or solitary metastatic lung cancer …”

P2: “Patients who were medically inoperable or who refused surgery for primary lung cancer or solitary metastatic lung cancer with diameters of 5.0 cm or less and were indicated for SBRT were candidates” => “Candidates were patients who were medically inoperable or who refused surgery for primary lung cancer or solitary metastatic lung cancer with diameters of 5.0 cm or less and were indicated for SBRT”

P3: Regarding “For primary lung cancers with diameters of 3 cm or less, … For primary lung cancers with diameters of 3.1-5.0 cm or metastatic lung cancer, …“ clarify metastatic cancer was always larger than 3 cm

P3: Regarding “… SRS was compared with multi-fraction SBRT…” more information about the multi-fraction SBRT is necessary such as dose and fraction sizes.

P4: “… but 1 patient have withdrew from …” => “… but 1 patient was withdrawn from …”

Figure 3: In (A) and (C), “… of local control after …” => “… of local recurrence after …”

P8: “… whoreceived …” => “who received …”

P8: In “As a result, the mean internal GTV doses in the 28-Gy arm and 30-Gy arm were 35.5 Gy and 40.1 Gy, respectively” aren’t they median of mean values?  Clearer description is necessary since it can be misleading.

P8: “… follow-up was period needed …” => “… follow-up period was needed …”

P8: “… BED3 value were 75.8 Gy …” => “… BED3 values were 75.8 Gy …”

Author Response

Response for Reviewer 3

We are grateful for your comments and advices. We revised our manuscript and your advices enabled us to improve the quality of our manuscript.

1). Please describe in detail what the risk-adapted strategy was and rationale underneath it.

Response:

Thank you for your advice. The explanation of the risk-adapted strategy was completely lacking. We modified and added following sentence to introduction section:

The risk-adapted strategy was that SRS doses were changed according to the risks of local recurrence. Patients were divided into two risk groups based on previous results of predictive factor analyses for local control [10]. A normal risk group was primary lung cancers with diameters of 3 cm or less and a high risk group was primary lung cancers with diameters of 3.1-5.0 cm or metastatic lung cancer regardless of tumor diameter. Radiation dose for the normal risk group was 28 Gy which was determined by the results of a phase I SRS-VMAT study and was almost the same biological effective dose (BED10) as 48 Gy in 4 fractions [11]. Radiation dose for the high risk group was increased up to 30 Gy [7].

(Ref no.7 showed that 1-year local recurrence rates were 13.8% in patients who received 34-Gy SRS and 2.0% in patients who received 30-Gy SRS)

2) There were only 3 patients for 30 Gy prescription compared to 18 patients for 28 Gy prescription, is it reasonable to analyze all together?  Provide your justification and discussion about it.  In addition, you may want to talk about whether complicated cases were from high dose group or not.

Response:

Thank you for your comments. As you mentioned, this problem is a major limitation of this study. We should have consulted statisticians before we created the protocol. We think that we will use the normal risk and high risk as a stratification factor for future controlled trial. In any case, we cannot justify this issue in this manuscript, therefore, we politely described this issue in the limitation section. Complicated cases according to dose level also added result section.

“Furthermore, 30-Gy SRS arm was only 3 patients compared to 18 patients for 28-Gy SRS arm. This is the problem of study design, therefore, reasonable study design is needed. Stratification according to the risks might be useful [31].”

Specific Comments  

P2: Regarding “Primary lung cancers with diameters of 3 cm or less were treated with 28-Gy SRS and primary lung cancers with diameters of 3.1-5.0 cm or metastatic lung cancer were treated with 30-Gy SRS” was it for “primary” only? This conflicts with “Patients who were medically inoperable or who refused surgery for primary lung cancer or solitary metastatic lung cancer …”

Response:

Thank you for your advice. We modified the sentences described as above.

P3: Regarding “For primary lung cancers with diameters of 3 cm or less, … For primary lung cancers with diameters of 3.1-5.0 cm or metastatic lung cancer, …“ clarify metastatic cancer was always larger than 3 cm

Response:

Thank you for your question. Metastatic cancer was regarded as high risk of local recurrence, therefore, 30 Gy was prescribed for metastatic lung cancer regardless of tumor diameter. To avoid confusion, we modified the sentence as follows:

For primary lung cancers with diameters of 3.1-5.0 cm or metastatic lung cancer regardless of diameter, 30 Gy in 1 fraction was prescribed to D95.

P3: Regarding “… SRS was compared with multi-fraction SBRT…” more information about the multi-fraction SBRT is necessary such as dose and fraction sizes.

Response:

Thank you for your advice. We added following sentence in the manuscript:

Multi-fraction SBRT consisted of 4- and 8-fraction SBRT, and 48 Gy in 4 fractions or 60 Gy in 8 fractions were typically prescribed to D95.

Thank you very much for other your specific comments. We modified what you pointed out. We thank you for your thorough review.